# An improved animal model for herpesvirus encephalitis in humans

Julia Sehl[1,2], Julia E. Hölper[1], Barbara G. Klupp[1], Christina Baumbach[3], Jens P. Teifke[2], Thomas C. Mettenleiter[1]*

1 Institute of Molecular Virology and Cell Biology, Friedrich-Loeffler-Institut, Greifswald-Insel Riems, Germany, 2 Department of Experimental Animal Facilities and Biorisk Management, Friedrich-Loeffler-Institut, Greifswald-Insel Riems, Germany, 3 Department of Animal Health Diagnostics, Food Safety and Fishery in Mecklenburg-Western Pomerania, Rostock, Germany

* thomas.mettenleiter@fli.de

**Data Availability Statement:** All relevant data are within the manuscript and Supporting Information files.

**Funding:** This study was supported by a predoctoral grant given to Julia Sehl by

## Abstract

Herpesviral encephalitis caused by Herpes Simplex Virus 1 (HSV-1) is one of the most devastating diseases in humans. Patients present with fever, mental status changes or seizures and when untreated, sequelae can be fatal. Herpes Simplex Encephalitis (HSE) is characterized by mainly unilateral necrotizing inflammation effacing the frontal and mesiotemporal lobes with rare involvement of the brainstem. HSV-1 is hypothesized to invade the CNS via the trigeminal or olfactory nerve, but viral tropism and the exact route of infection remain unclear. Several mouse models for HSE have been developed, but they mimic natural infection only inadequately. The porcine alphaherpesvirus Pseudorabies virus (PrV) is closely related to HSV-1 and Varicella Zoster Virus (VZV). While pigs can control productive infection, it is lethal in other susceptible animals associated with severe pruritus leading to automutilation. Here, we describe the first mutant PrV establishing productive infection in mice that the animals are able to control. After intranasal inoculation with a PrV mutant lacking tegument protein pUL21 and pUS3 kinase activity (PrV-ΔUL21/US3Δkin), nearly all mice survived despite extensive infection of the central nervous system. Neuroinvasion mainly occurred along the trigeminal pathway. Whereas trigeminal first and second order neurons and autonomic ganglia were positive early after intranasal infection, PrV-specific antigen was mainly detectable in the frontal, mesiotemporal and parietal lobes at later times, accompanied by a long lasting lymphohistiocytic meningoencephalitis. Despite this extensive infection, mice showed only mild to moderate clinical signs, developed alopecic skin lesions, or remained asymptomatic. Interestingly, most mice exhibited abnormalities in behavior and activity levels including slow movements, akinesia and stargazing. In summary, clinical signs, distribution of viral antigen and inflammatory pattern show striking analogies to human encephalitis caused by HSV-1 or VZV not observed in other animal models of disease.

## Author summary

In developed countries, more than 50% of humans are seropositive for the neurotropic Herpes Simplex Virus 1 (HSV-1) and two to four million cases of Herpes simplex

Studienstiftung des deutschen Volkes. The funder did not play any role in the study design, data collection and analysis, decision to publish or preparation of the manuscript.

**Competing interests:** The authors have declared that no competing interests exist.

encephalitis (HSE) are reported per year worldwide. Primary infection with HSV-1 takes place via the skin or the oral mucosa followed by intraaxonal retrograde spread to sensory ganglia of the peripheral nervous system where HSV-1 usually establishes latency. Further spread to the central nervous system results in HSE, a necrotizing encephalitis effacing predominantly the temporal and frontal lobes of the brain. Mice infected with HSV-1 develop encephalitis, but do not show the typical lesions and exhibit high mortality rates. Here we demonstrate that mice infected with a mutant pseudorabies virus lacking the tegument protein pUL21 and an active viral kinase pUS3 were able to survive the productive infection but developed lymphohistiocytic encephalitis with viral antigen distribution, inflammation and associated behavioral changes comparable to HSE in humans. These striking analogies offer new perspectives to study herpesviral encephalitis in a suitable animal model.

## Introduction

Herpes Simplex Virus-1 (HSV-1) infection is one of the most frequent causes of necrotizing encephalitis (HSE) in humans [1]. HSE is associated either with primary HSV-1 infection occurring mostly in children and adolescents, or reactivation of latent HSV-1 in adults. Mortality rates up to 70% have been reported when untreated [2–4]. Patients suffering from HSE mostly show fever, headache, lethargy, aphasia, disorientation, behavioral changes and seizures [2, 5]. Sequelae can include neurological deficits and dysfunctions such as behavioral and cognitive abnormalities, memory impairment, as well as seizures [1, 5, 6]. HSE mainly manifests in the frontal and mesiotemporal lobes including the limbic system, usually asymmetrically with relatively rare encephalitis in the brainstem [7]. Histopathologically, HSE is characterized by neuronal necrosis, mononuclear infiltrates and microgliosis [8, 9].

Varicella-Zoster Virus (VZV) is another neurotropic human pathogen, which causes Varicella (chickenpox) in young children after primary infection or herpes zoster (rash, shingles) primarily in elderly individuals [2, 10]. After primary infection, VZV establishes latency in dorsal root ganglia where it can reactivate from, and cause vesicular skin eruptions typically located unilaterally within a dermatome [11]. The rash can migrate to neighboring dermatomes and is associated with pain and extreme itching [10]. VZV can also affect other organs causing severe central nervous system (CNS) manifestations [12], but in contrast to HSV-1, vasculopathy is the key pathologic hallmark [13]. Meningoencephalitis occurs either after Varicella or Herpes Zoster in immunocompetent or immunosuppressed patients, representing as disseminated encephalomyelitis or inflammation with an unspecific distribution pattern [10]. As in HSV-1, clinical signs include alteration in mental status and focal neurological deficits, whereas the development of seizures is rare [14–16].

The herpesviruses HSV-1, VZV as well as Pseudorabies Virus (PrV) belong to the subfamily of the *Alphaherpesvirinae*. PrV is the causative agent of Aujeszky's disease in pigs but infects a variety of mammals ranging from ruminants to carnivores, rodents and lagomorphs. Horses and primates resist infection [17], although evidence for rare human infections have been reported recently [18, 19]. In adult pigs, the virus causes mostly respiratory illness, abortion or even subclinical infection, whereas piglets usually develop severe neurological signs. However, only pigs can survive a productive infection, while all other susceptible animals develop severe neurological signs and extreme pruritus, known as "mad-itch" syndrome, and succumb to death within a short time after infection [17].

In the past, different mechanisms have been discussed how alphaherpesviruses gain access to the CNS in humans [7]. Several mouse models with different age, genetic background,

inoculation routes, and virus strains of variable neurovirulence have been developed to investigate the pathogenesis of viral infections *in vivo* [20]. So far, mostly intranasal or intracerebral infection models have been used to study HSE in mice [21–23], whereas HSV-1 latency is established mainly by corneal scarification [21]. In general, after replication in the nasal mucosa and the trigeminal ganglion, HSV-1 has been detected in the brainstem (BS), cerebellum, thalamus (TH), hippocampus (HIP) or lateral ventricles, thus establishing a more diffuse infection pattern than observed in HSE patients [24]. Although viral antigen was present in the olfactory bulb, the virus did not spread to the temporal and frontal lobes, which are the typical locations in HSE in humans [22, 25]. Thus, the available models mimic HSE only insufficiently limiting experimental studies on the short- as well as long-term consequences of herpesviral infection.

Murine infection models of PrV have shown that the virus enters peripheral sensory neurons through free nerve endings. Via retrograde intraaxonal spread the virus is able to invade the CNS [26]. As shown in Fig 1, after intranasal infection, which mimics a natural route, PrV replicates initially in the respiratory epithelium of the nose (RE) [27–29]. The virus then travels along afferent trigeminal sensory fibers to neurons of the trigeminal ganglion (TG), an invasion route also used by HSV-1 [30]. First order neurons of the TG reach into the CNS through the pons located in the metencephalon (MET). These neurons are synaptically connected to three trigeminal nuclei present in the myelencephalon (spinal trigeminal nucleus, Sp5), MET (principal sensory trigeminal nucleus, Pr5) and mesencephalon (MES) (mesencephalic trigeminal nucleus, Me5). Second order neurons from the trigeminal nuclei project to the ventral posteromedial nucleus of the TH [31] whereas third order neurons originating from TH eventually reach the primary somatosensory cortex (SSC) in the parietal lobe (PL) [32]. For PrV, invasion of the CNS is also facilitated through sympathetic and parasympathetic nerve endings including their autonomic ganglia as well as through the facial nerve [27–29].

Since PrV also invades the CNS of pigs via the olfactory route [33], this alternative way of infection may also occur in mice and humans infected with HSV-1 [2, 7, 34–38], but still requires more in-depth investigation.

Intranasal PrV infection of mice resulted in a fulminant disease with animals succumbing 2–3 days after wild-type virus infection. At this stage, there is an abundance of infected neurons in the TG and trigeminal brainstem nuclei. Infection studies with a series of PrV deletion mutants resulted in extended survival times correlating with virus intrusion into higher areas of the CNS including the cerebral cortex [28, 29]. However, productive infection was always 100% fatal.

In contrast, a PrV mutant simultaneously deleted in genes encoding pUL21 and pUS3 was the only virus in our studies leading to a productive infection in mice, which the animals were able to control. Therefore we investigated in detail neuroinvasion of this virus as well as the inflammatory response to infection after construction of a novel mutant virus, PrV-ΔUL21/US3Δkin, which lacked the UL21 gene but contained only a specific mutation in pUS3 inactivating its kinase function.

The mean time to death (MTD), the clinical phenotype as well as kinetics of viral spread and inflammatory responses were compared to wildtype PrV strain Kaplan (PrV-Ka) and single mutant virus infected mice.

## Results

### In vitro replication properties

Replication of PrV-ΔUL21/US3Δkin was analyzed in rabbit kidney (RK13) cells and compared to infection with PrV wildtype strain Kaplan (PrV-Ka) and the single mutants PrV-ΔUL21

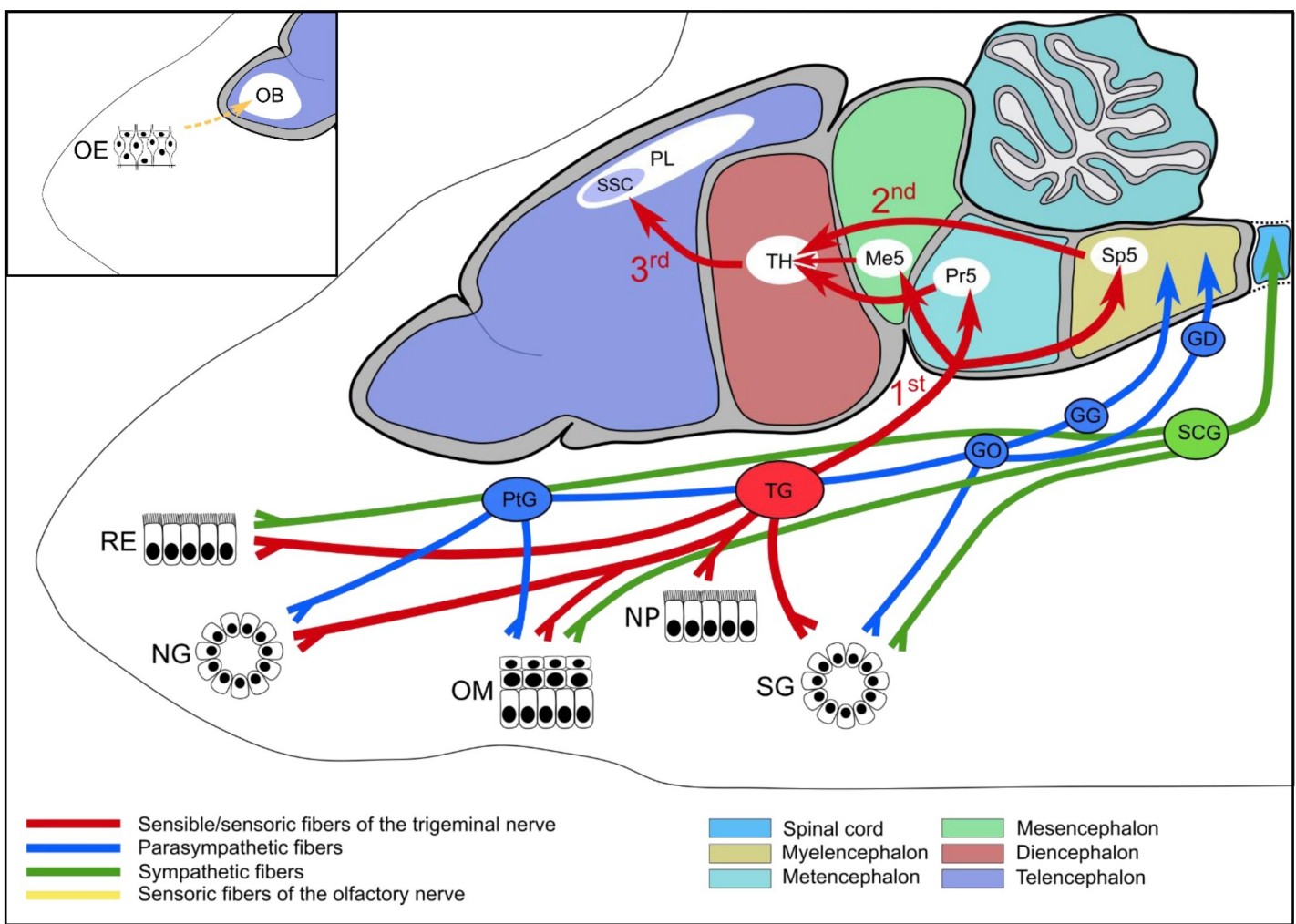

**Fig 1. Schematic illustration of viral spread of PrV in mice after intranasal infection.** After initial replication in the respiratory epithelium (RE) PrV primarily invades sensible/sensoric fibers of the trigeminal nerve (red) and is transported via 1st order neurons to trigeminal nuclei (Sp5, Pr5, Me5). From there, 2nd order neurons project to the thalamus (TH). Eventually, thalamic fibers end up as third order neurons in the somatosensory cortex (SSC) located in the parietal lobe (PL). PrV does also infect the nasal glands (NG), oral mucosa (OM), nasopharynx (NP), and salivary glands (SG) which are all innervated by the trigeminal nerve. Viral spread further occurs via sympathetic (green) or parasympathetic fibers (blue) and the corresponding autonomic ganglia such as the superior cervical ganglion (SCG) and pterygopalatine ganglion (PtG), otic ganglion (GO), geniculate ganglion (GG), and distal ganglion (GD), respectively. Although not yet reported in mice, herpesviral infection may be facilitated through the olfactory epithelium (OE) and the olfactory nerve (yellow) to finally infect the olfactory bulb (OB) (insert).

[39] and PrV-US3Δkin [40]. Cells were infected at a MOI of 3, and harvested after 24 h. Progeny virus titers were determined on RK13 cells. As shown in Fig 2A, titers of the single mutants were approx. 5- to 10-fold lower compared to PrV-Ka as described earlier [39, 40], while PrV-ΔUL21/US3Δkin showed a more drastic effect with a 50- to 100-fold titer reduction.

Plaque diameters as a measure for cell-to-cell spread capability were determined 48 h after infection (Fig 2B). Plaque diameters reached by PrV-Ka were set as 100% and values of the single and double mutants were calculated accordingly. PrV-US3Δkin showed only slightly reduced plaque sizes as was shown for mutants lacking pUS3 completely (PrV-ΔUS3; [40] while PrV-ΔUL21 reached only 60% plaque size [39]). Only 50% plaque diameter was reached with PrV-ΔUL21/US3Δkin pointing to an additive effect of both mutations. However, despite the effects on viral replication and plaque sizes, PrV-ΔUL21/US3Δkin showed productive replication.

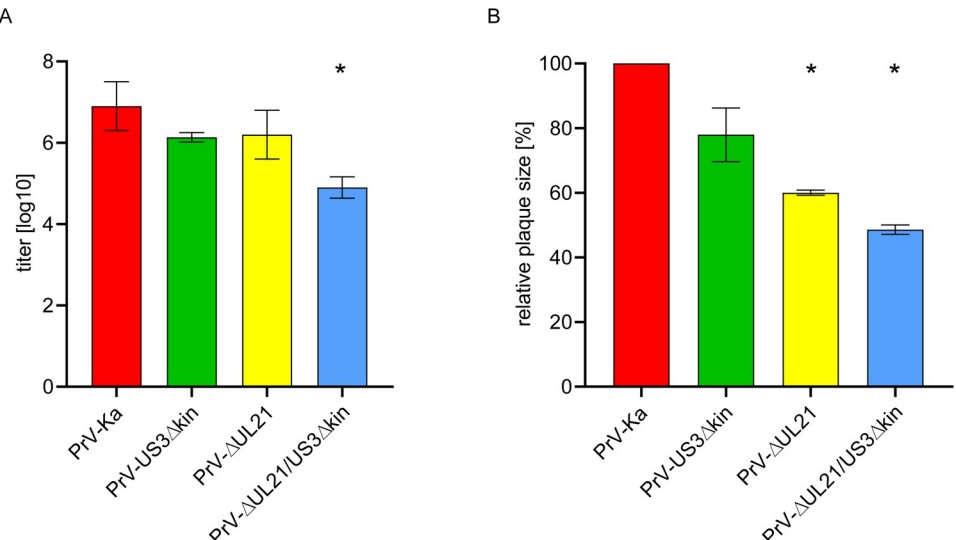

**Fig 2. In vitro growth properties and cell-to-cell spread of mutant viruses.** (A) RK13 cells were infected with PrV-Ka and the respective virus mutants at a MOI of 3, and harvested 24 h p.i. Plaque assays performed on RK13 cells were used to determine the titer of virus progeny. (B) Plaque diameters of the different virus mutants were assessed on RK13 cells. 20 plaques each per virus mutant were measured and compared to plaque diameter of PrV-Ka set as 100%. Average values and standard deviations of three independent experiments are shown. Significant differences of virus mutants compared to PrV-Ka are indicated by an asterisk (*).

## Mean time to death

To assess the MTD mice were infected intranasally with 1 x $10^4$ PFU of PrV-ΔUL21/US3Δkin (n = 6) or PrV-Ka (n = 4), PrV-US3Δkin (n = 4), PrV-ΔUL21 (n = 4), and supernatant of non-infected cells (n = 2) as control. The animals were monitored daily, scored according to the scale given in S1 Table, and euthanized when the humane endpoint was reached. Relative survival rates are shown in Fig 3. On average, mice infected with PrV-Ka and PrV-US3Δkin reached the humane endpoint at 62 h p.i. and 63 h p.i., respectively (S2 Table). With 110 h, the MTD of mice infected with PrV-ΔUL21 was almost twice as long as in PrV-Ka-infected animals, but all animals succumbed to the infection. Mice inoculated with PrV-ΔUL21/US3Δkin, however, survived until the end of the experiment (450 h p.i.), except for one animal which had to be euthanized at 239 h p.i. (Fig 3; S2 Table).

## Clinical evaluation

First clinical signs appeared at 47 and 51 h p.i. in PrV-Ka and PrV-US3Δkin-infected mice, respectively (S2 Table). PrV-ΔUL21-infected animals showed a delay in the onset of clinical signs starting 96 h p.i., and clinical signs developed slower but similar to PrV-Ka infection e.g. ruffled fur, hunched posture and mild pruritus. Subsequently, clinical signs aggravated and included mainly unilateral conjunctivitis, nasal bridge edema and multiple facial hemorrhagic skin erosions and ulcerations due to excessive pruritus and beginning auto mutilation. Mice were apathetic and showed phases of intermittent hyperactivity. In the final stage, some animals became dyspneic. These observations reflect previous data [28, 29], although mice in general survived slightly longer in the present study.

After infection with PrV-ΔUL21/US3Δkin, mice began to develop first clinical signs at 152 h p.i. on average. The animals showed mild pruritus starting at the head, but spreading to the whole body in the following days, particularly to the flank. This was further accompanied by

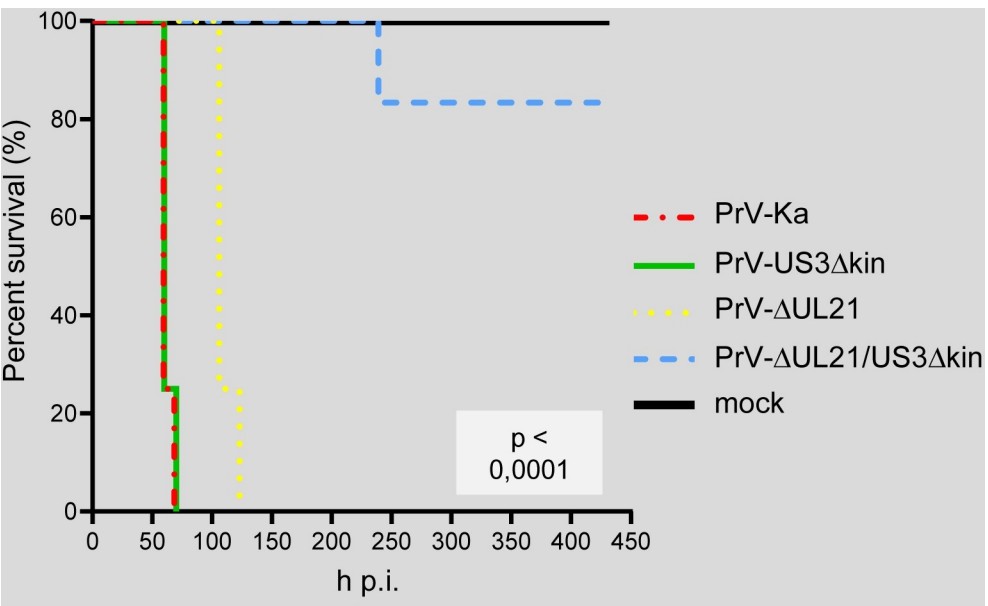

**Fig 3. Kaplan-Meier curve showing the relative survival rates of mice after inoculation with PrV-Ka and mutants.** Mice (PrV-Ka, n = 4; PrV-US3Δkin, n = 4; PrV-ΔUL21, n = 4; PrV-ΔUL21/US3Δkin, n = 6) were inoculated intranasally with a total of 10μl virus suspension containing $1x10^4$ PFU. The animals were daily monitored and sacrificed in the moribund stage. In contrast to mice infected with PrV-Ka, PrV-US3Δkin and PrV-ΔUL21, PrV-ΔUL21/US3Δkin-infected animals survived the infection except for one animal that had to be euthanized at 239 h p.i. The survival rates differ significantly (log-rank test, p < 0,0001) between the groups.

phases of decelerated movements, retarded reactions to external stimuli, and reduced activity levels followed by phases of normal behavior. Moreover, signs of photophobia were present in individual animals. Of note, clinical signs reached their climax between 216 and 312 h after infection. As mentioned earlier, within this time one mouse had to be euthanized due to severe itching and bad condition. All other mice showed only mild to moderate clinical signs such as hunching, ruffled fur or even mild pruritus, until they recovered and survived the infection. Unlike PrV-Ka and single mutant infections, hemorrhagic erosions never appeared (S2 Table).

### Kinetics of neuroinvasion

To investigate the clinical phenotype as well as the kinetics of viral neuroinvasion and inflammatory reaction in more detail, three mice each were inoculated with the respective viruses and sacrificed at different time points after infection (Fig 5). As negative controls, mock-infected mice were euthanized after 24, 168, 312 and 504 h.

**Clinical phenotype.** Prevalence and distribution of clinical scores of PrV-ΔUL21/US3Δkin infected animals compared to PrV-Ka and single mutant infected mice are presented in Fig 4. As observed previously, in PrV-Ka and PrV-US3Δkin infected animals clinical signs drastically increased very early after infection, although with a delay in PrV-ΔUL21 infected mice. Most animals showed severe clinical signs at the endpoint (Fig 4). In contrast, PrV-ΔUL21/US3Δkin-infected mice developed first mild clinical signs starting 144 h p.i. with a subsequent increase of the number of sick animals (Fig 4D). Most mice showed only moderate clinical signs including slowed movements and/or reduced reactions to external stimuli up to total inactivity with periods of akinesia and stargazing. Foci of alopecia mainly in the face, but also in other body regions, e.g. the neck, limbs, abdomen, back, flank and tail root occurred. Clinical signs reached their climax between 192 and 360 h p.i. Within this time period, some

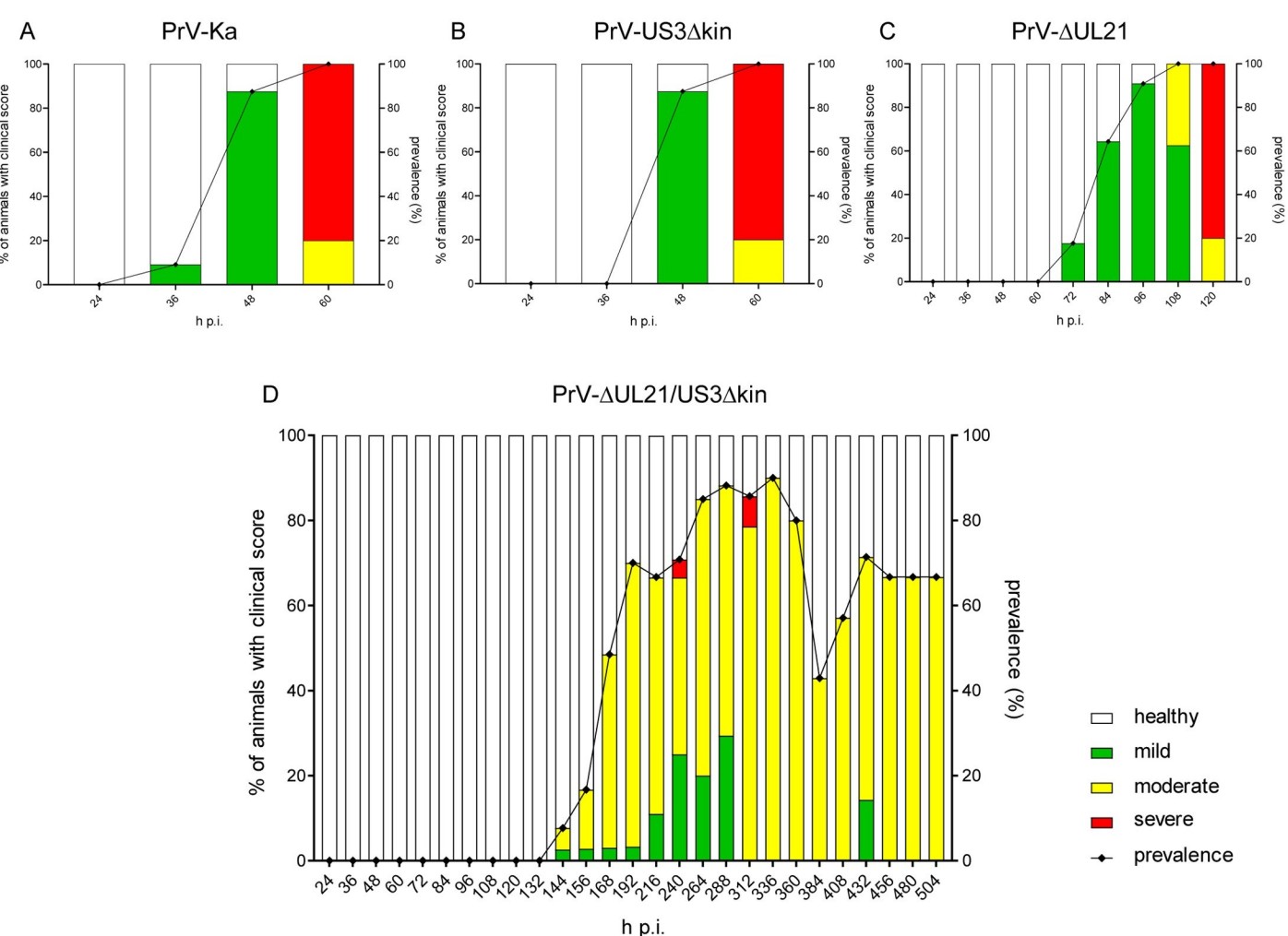

**Fig 4. Prevalence of diseased animals after infection with PrV-Ka, PrV-US3Δkin, PrV-ΔUL21 and PrV-ΔUL21/US3Δkin in the kinetic trial.** Starting 24 h p.i. at least 3 mice per indicated time point were sacrificed and submitted for pathohistological investigation. The frequency of sick animals infected with PrV-Ka (A), PrV-US3Δkin (B), PrV-ΔUL21 (C) or PrV-ΔUL21/US3Δkin (D) per time is indicated by bars. Based on a scoring system (S1 Table) animals were categorized into either mildly (green), moderately (yellow) or severely affected (red). Prevalence is given by the black line on the right Y-axis.

animals were moderately or even severely affected. One mouse had to be euthanized 240 h p.i. and another one died unexpectedly at 301 h p.i., but the other animals survived beyond this point. Thus, the period between 216 and 336 h p.i. is the critical phase of PrV-ΔUL21/ US3Δkin-infection, where the animals either exhibit only mild to moderate clinical signs leading to survival, or their condition deteriorated to a level requiring euthanasia or resulting in death.

**Neuroinvasion and kinetics of viral spread.** To track viral infection of the brain, 16 head sections per animal were analyzed by immunohistochemistry using an anti-PrV glycoprotein B (α-gB) serum [109].

In animals infected with PrV-Ka, viral antigen was first detected at 36 h p.i., and significantly increased from 48 to 60 h p.i. (Fig 5A). As shown in Fig 6 TG, second order neurons (Sp5), autonomic ganglia, nasal and oral epithelium and the medullary formatio reticularis (FR) with rostroventrolateral reticular nucleus (RVL) showed massive antigen staining. Compared to PrV-Ka, PrV-US3Δkin infection showed a slightly delayed spread but with comparable distribution of PrV antigen. At the final stage of PrV-Ka and PrV-US3Δkin infection (60 h

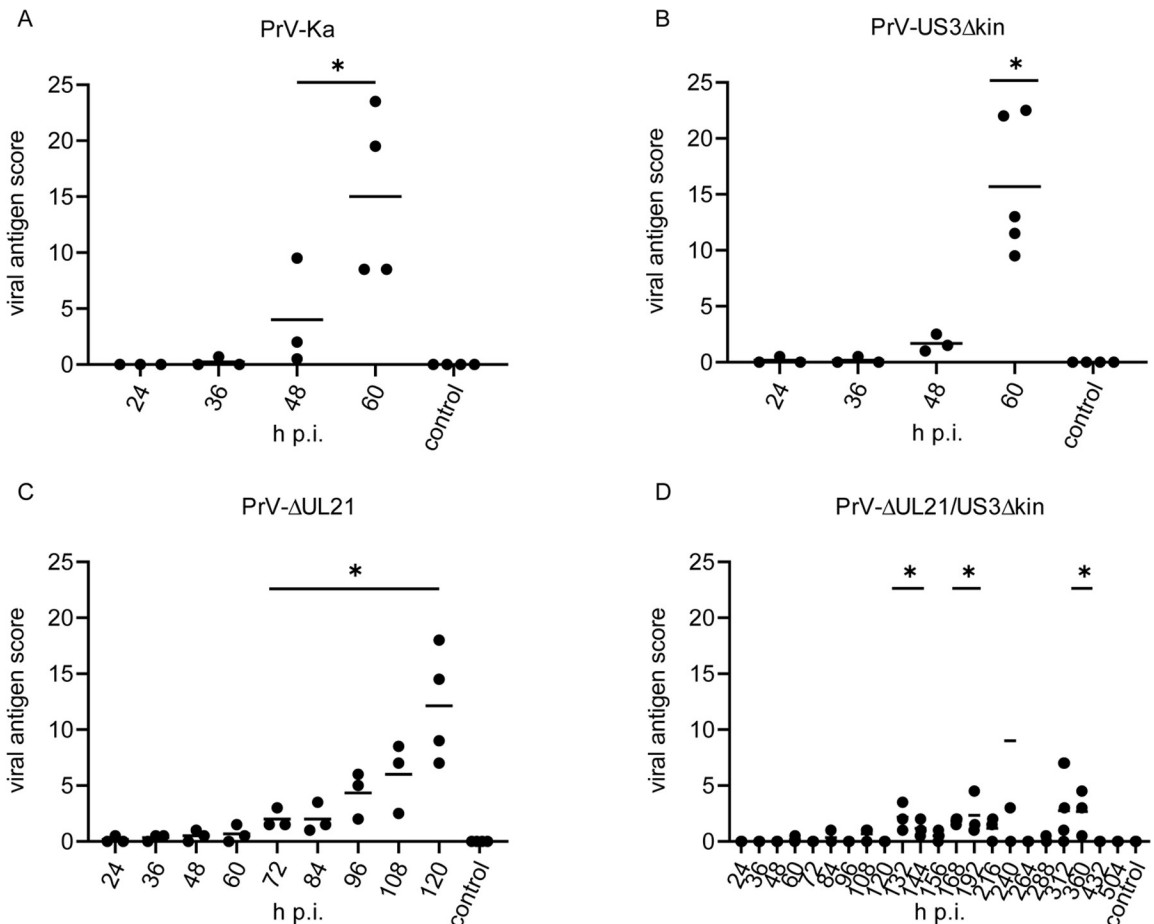

**Fig 5. Viral antigen score in mice infected with mutant viruses.** Histological specimen of mice infected with PrV-Ka (A), PrV-US3Δkin (B), PrV-ΔUL21 (C) and PrV-ΔUL21/US3Δkin (D) were scored for viral antigen distribution with negative (0), focal to oligofocal (1), multifocal (2) and diffuse (3). The final average value resulted from points given for each tissue section analyzed. Significant differences of mutant viruses compared to the mock-infected control group are indicated with an asterisk (*).

p.i.), animals showed the highest viral antigen score (Fig 5A and 5B). PrV antigen was also found in different parts of the BS including the solitary tract (Sol) and FR with RVL, lateral reticular nucleus (LRt), parvicellular reticular nucleus (PCRt) and medullary reticular nucleus (dorsal part, MdD). The brain including the mes- (MES), di- (DI) and telencephalon (TEL), however, was free of viral antigen in both PrV-Ka and PrV-US3Δkin-infected mice. Details are given in S1 Fig. Although viral spread was consistent with previous studies [28, 29], infection of the FR had not been described yet. In PrV-ΔUL21-infected animals, viral antigen score increased more slowly compared to PrV-Ka-infected mice (Fig 5C). As shown in Fig 6, the distribution of viral antigen was similar compared to PrV-Ka and PrV-US3Δkin, but was in addition present in a few neurons of the inferior olive (IO), hypothalamus (HYPO) including the lateral hypothalamic area (LH), TH with ventromedial thalamic nucleus (VM) and mesiodorsal thalamic nucleus (lateral part), frontal lobe (FL) with agranular insular cortex (AI), parietal lobe (PL) with secondary auditory cortex (ventral area, AuV) and secondary somatosensory cortex (S2) and piriform lobe (PIR) at the time of euthanasia, indicating that longer survival results in more extensive neuronal spread towards the cerebral cortex.

In PrV-ΔUL21/US3Δkin-inoculated animals, viral antigen was detectable at varying amounts over the study period. The number of viral antigen positive cells was lower than in

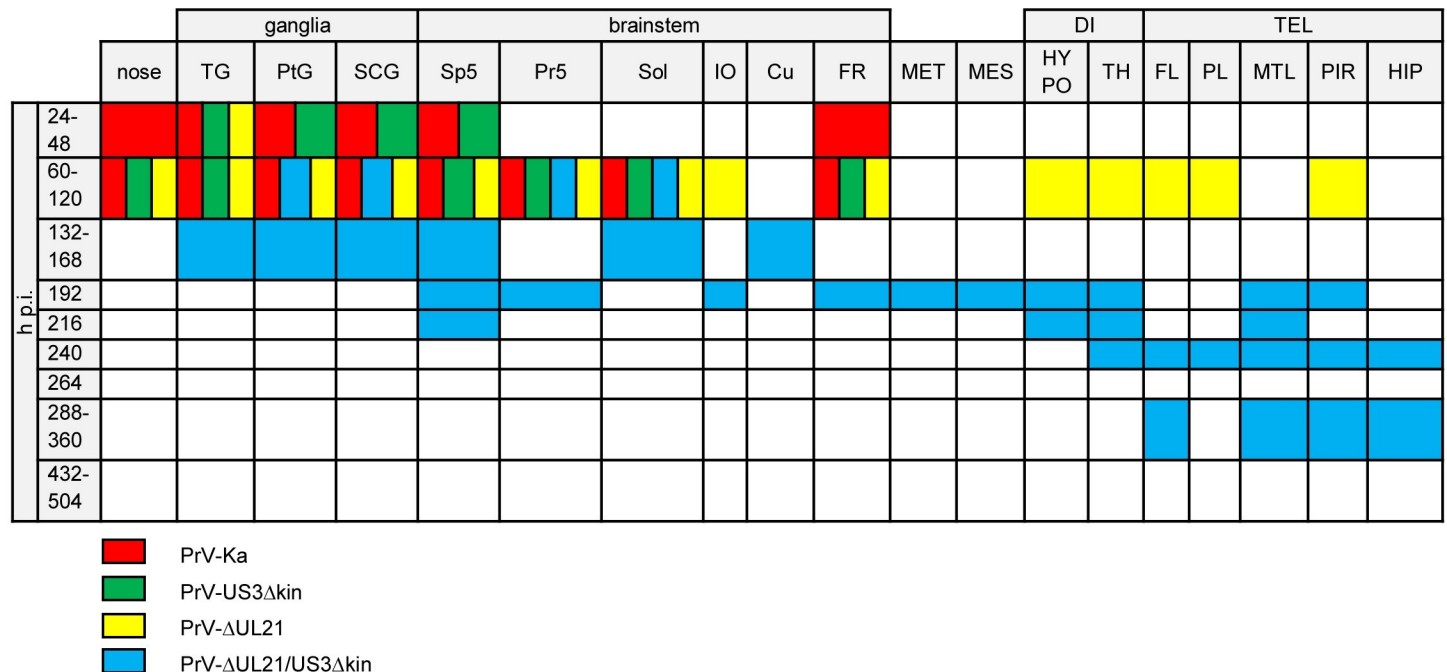

**Fig 6. Antigen distribution.** Colored boxes indicate viral antigen in the respective areas after infection with PrV-Ka (red), PrV-US3Δkin (green), PrV-ΔUL21 (yellow) and PrV-ΔUL21/US3Δkin (blue). TG = trigeminal ganglion, PtG = pterygopalatine ganglion, SCG = superior cervical ganglion, Sp5 = spinal trigeminal nucleus, Pr5 = principal sensory trigeminal nucleus, Sol = solitary tract, IO = inferior olive, Cu = cuneate nucleus, FR = reticular formation, MET = metencephalic regions others than Pr5, MES = mesencephalic regions others than Me5, DI = diencephalic regions, HYPO = hypothalamus, TH = thalamus, TEL = telencephalic regions, FL = frontal lobe, PL = parietal lobe, MTL = mesiotemporal lobe, PIR = piriform lobe, HIP = hippocampus.

animals infected with PrV-Ka or single mutants (Fig 5D). The distribution of viral antigen is shown in Fig 6. Between 60 and 120 h p.i., as in PrV-Ka and single mutant infected mice, autonomic ganglia and parts of the BS including trigeminal second order neurons (Pr5) and Sol were positive. Between 132 to 168 h p.i. additionally TG, Sp5 and cuneate nucleus (Cu) became infected. The BS including IO and FR with PCRt and medullary reticular nulceus (ventral part) was antigen-positive at 192 h p.i. Viral antigen was further detectable in single neurons of the MET including the lateral parabrachial nucleus (central part, LPBC), in MES with lateral periaqueductal grey (LPAG), in TH, in HYPO with posterior hypothalamic area (PH) as well as in the TEL including the mesiotemporal lobe (MTL) with ectorhinal cortex (ECT) and lateral entorhinal cortex (LENT) and PIR. At 216 h p.i. viral antigen was detectable in second order neurons (Sp5), in HYPO with LH, in TH with dorsal lateral geniculate nucleus, and in MTL with LENT, the latter showing an increasing amount of PrV-gB positive neurons. At 240 h p.i., one animal had to be euthanized due to severe clinical signs. Histopathologically, this mouse showed a marked infection almost throughout the FL with primary motor cortex (M1), secondary motor cortex (M2), agranular insular cortex (posterior part, AIP), agranular insular cortex (dorsal part, AID), agranular insular cortex (ventral part, AIV), granular insular cortex (GI), dysgranular insular cortex (DIC), PL with primary somatosensory cortex (S1) and secondary somatosensory cortex (S2), MTL with ECT, LENT, perihinal cortex (PRH), temporal association cortex (TeA) and hippocampus (HIP), while all other animals had a low viral antigen load at that time in the respective areas as well as in TH and PIR. In brain sections of mice sacrificed at 264 h p.i., no viral antigen was detectable. Similar to the animal which had to be euthanized at 240 h p.i., in the mouse that died between 288 and 312 h p.i. the virus invaded cortical areas. Surprisingly, in animals sacrificed at 288 and 360 h p.i., viral antigen was again

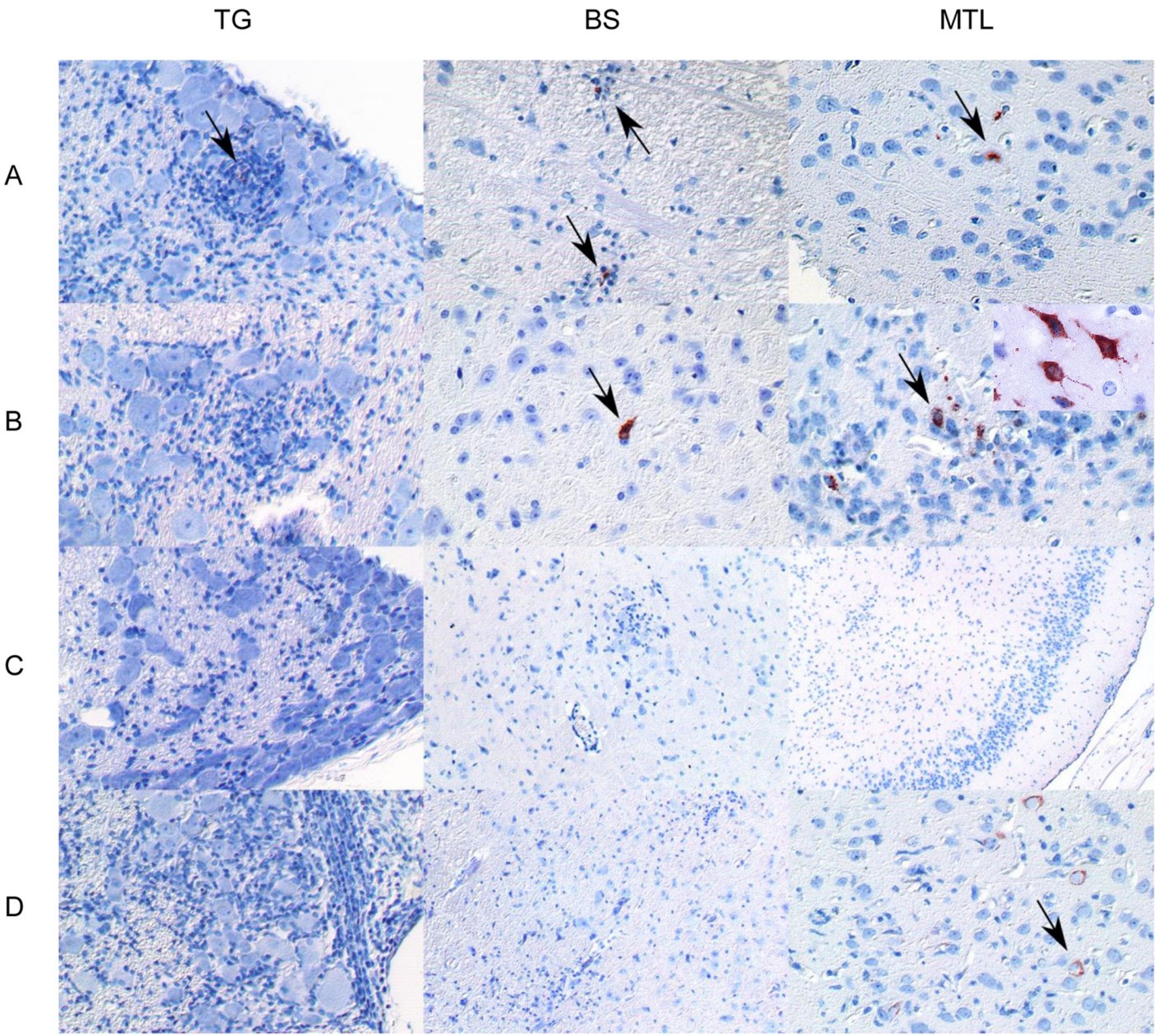

**Fig 7. Detection of viral antigen in mice infected with PrV-ΔUL21/US3Δkin.** Immunohistochemistry (anti PrV-gB) of the trigeminal ganglion (TG), brainstem (BS) and mesiotemporal lobe (MTL) are shown in A) 132–168 h p.i., B) 192–240 h p.i., C) 264 h p.i. and D) 288–360 h p.i. Positive staining is highlighted by arrows. A higher magnification of single infected neurons is shown in the inset in B) MTL.

detectable in FL with M2 and agranular insular cortex (AI), MTL with ECT, LENT and PRH, PIR and HIP. In brain sections of mice analyzed at later time points (432 and 504 h p.i.) no antigen was detectable (summarized in S2 Fig). Representative examples of immunohisto-chemical analyses of TG, BS and MTL of animals infected with PrV-ΔUL21/US3Δkin for different times are shown in Fig 7.

**Kinetics of the inflammatory response.** Head sections were stained with hematoxylin/eosin and histopathologically analyzed. Inflammatory scoring results of mice infected with

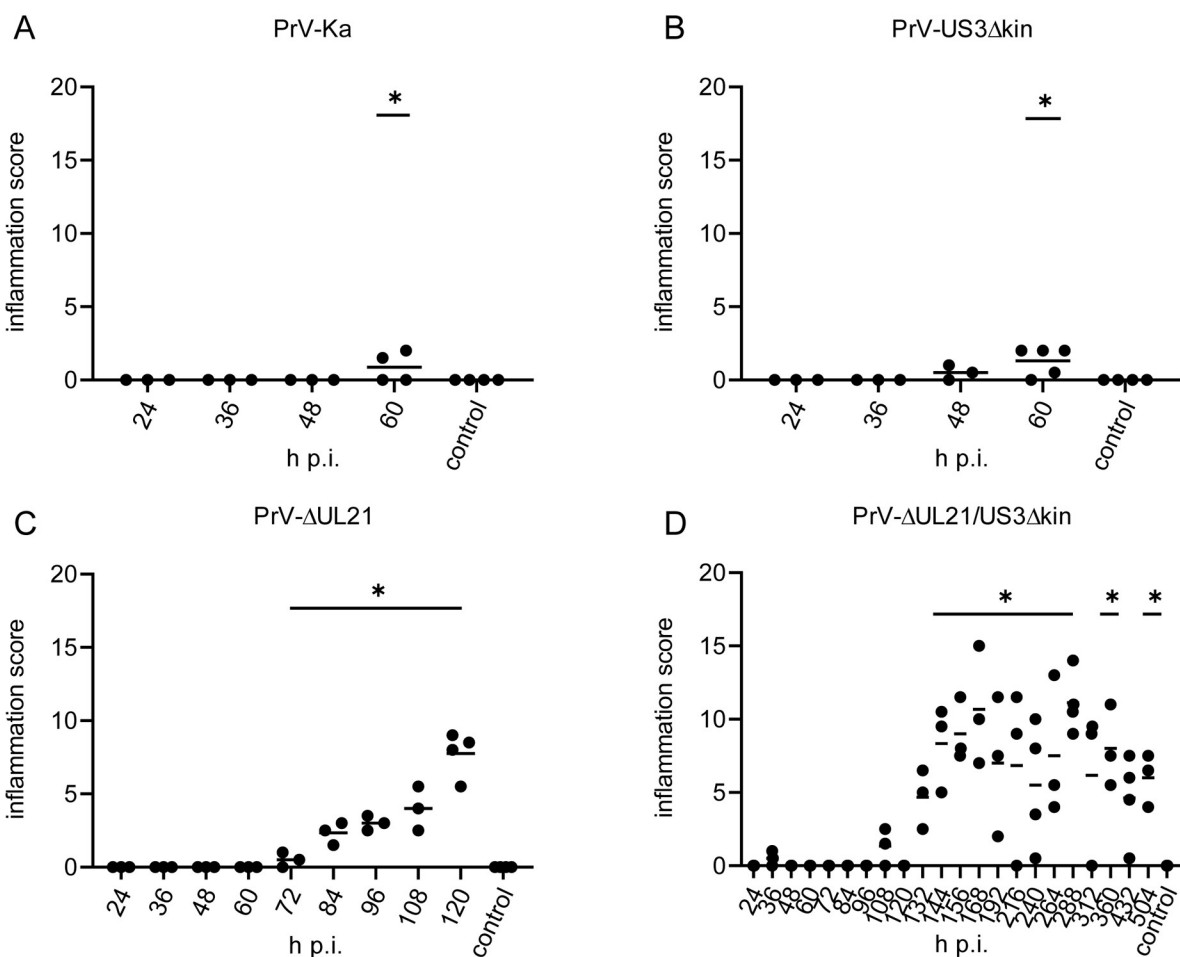

**Fig 8. Inflammation score in infected mice.** Histological specimen of mice infected with PrV-Ka (A), PrV-US3Δkin (B), PrV-ΔUL21 (C) and PrV-ΔUL21/US3Δkin (D) were scored for inflammation with 0 = no inflammation, 1 = mild inflammation, 2 = moderate inflammation, 3 = severe inflammation. The final average value resulted from points given for each tissue section analyzed. Significant differences of mutant viruses compared to the mock-infected control group are indicated with an asterisk (*).

PrV-Ka, PrV-US3Δkin, PrV-ΔUL21 and PrV-ΔUL21/US3Δkin are shown in Fig 8 and the distribution of inflammatory reaction is indicated in Fig 9.

Only limited inflammation was observed in mice infected with PrV-Ka (Fig 8A) and PrV-US3Δkin (Fig 8B). Signs of inflammation were only present in the nose and in superior cervical ganglion (SCG) (Fig 9). In contrast, inflammatory response was significantly higher in mice infected with PrV-ΔUL21 (Fig 8C). Within 72 to 108 h p.i. mild to moderate inflammation of ganglia was present (Fig 9). Very mild lymphohistiocytic brainstem encephalitis occurred starting at 108 h p.i. Inflammatory changes were not detectable in MES, MET, DI and TEL.

Mice infected with PrV-ΔUL21/US3Δkin showed the highest inflammation score (Fig 8D). Besides an immediate inflammatory reaction at 36 h p.i. in the nose, inflammation was not detectable until 96 h p.i. At 108 h p.i. inflammatory response was found in the TG and in SCG, and subsequently inflammation scores significantly increased from 132h p.i. until the termination of the experiment (Fig 8). As presented in Fig 9, very mild inflammation in the nose was observed starting 36 h p.i., whereas after 108 h p.i. mild ganglionitis and ganglioneuritis in TG and autonomic ganglia was found. At 120 h p.i., no signs of inflammation were detectable, but

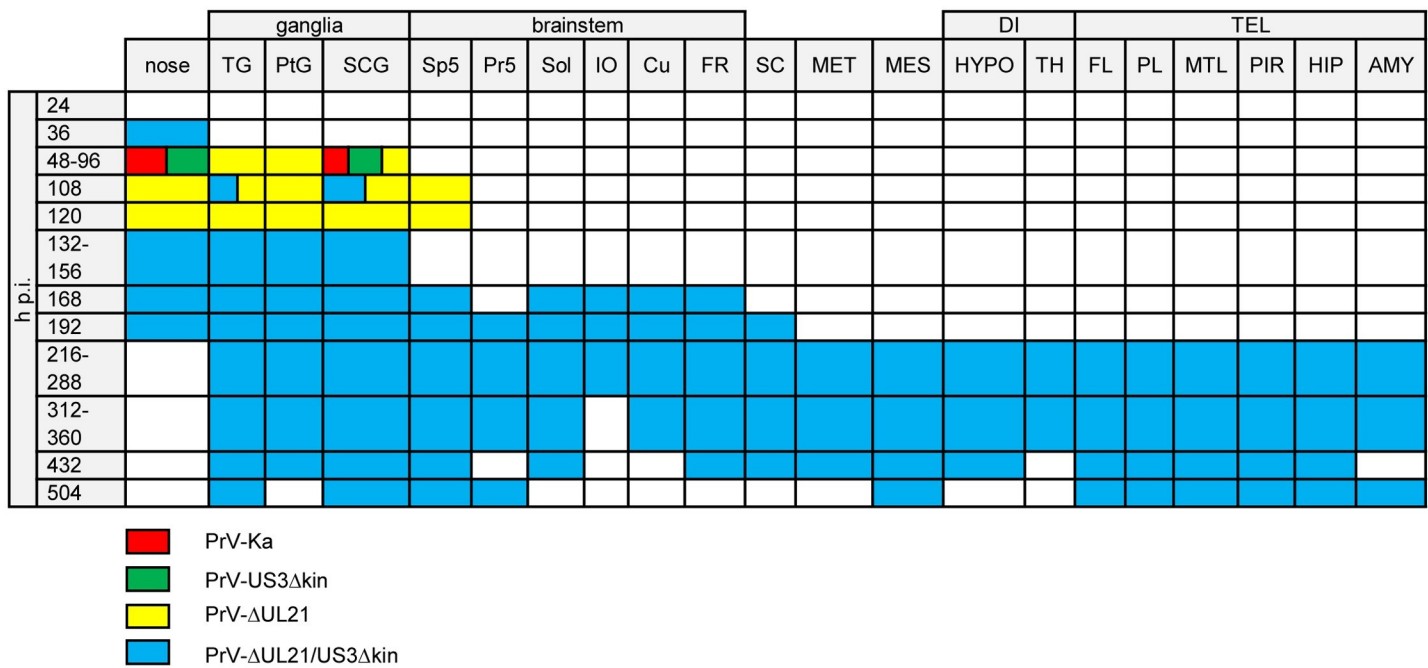

**Fig 9. Sites of inflammatory response.** Colored boxes indicate inflammation in the respective areas after infection with PrV-Ka (red), PrV-US3Δkin (green), PrV-ΔUL21 (yellow) and PrV-ΔUL21/US3Δkin (blue). TG = trigeminal ganglion, PtG = pterygopalatine ganglion, SCG = superior cervical ganglion, Sp5 = spinal trigeminal nucleus, Pr5 = principal sensory trigeminal nucleus, Sol = solitary tract, IO = inferior olive, CU = cuneate nucleus, FR = reticular formation, SC = spinal cord, MET = metencephalic regions others than Pr5, MES = mesencephalic regions others than Me5, DI = diencephalic regions, HYPO = hypothalamus, TH = thalamus, TEL = telencephalic regions, FL = frontal lobe, PL = parietal lobe, MTL = mesiotemporal lobe, PIR = piriform lobe, HIP = hippocampus, AMY = amygdala.

ganglionitis aggravated thereafter. At 168 h p.i. inflammation was found in the brainstem including second order neurons (Sp5), Sol, IO, Cu and RF with PCRt, RVL, MdD, lateral para-gigantocellular nucleus, intermediate reticular nucleus and LRt. At 192 h p.i. infiltrates were focally present in trigeminal second order neurons (Pr5) and in the cervical spinal cord. Between 216 and 240 h p.i., mild brainstem encephalitis turned to moderate and expanded to higher neuronal areas causing inflammation of the brain and meninges in MTL with LENT and PRH, PIR, amygdala (AMY) with posterolateral cortical amygdaloid nucleus as well as in MES with deep mesencephalic nucleus (DpMe) and ventral tegmental area (VTA), and in DI with substantia nigra reticular part, parasubthalamic nucleus, mamillothalamic tract, LH, PH, dorsomedial thalamic nucleus and ventromedial hypothalamic nucleus. At 264 to 288 h p.i, ganglia, BS and spinal cord were moderately affected, whereas MES with DpMe and DI with VM showed only few inflammatory infiltrates, but meningoencephalitis further spread to FL including AIP, M1, M2, lateral orbital cortex (LO), medial orbital cortex (MO), PL with S1, S2, and HIP. In animals investigated between 312 and 360 h p.i. meningoencephalitis progressed even further and included FL with AI, AIP, AID, LO, M1, M2, GI, DIC, AIV, LO, MO, ventral orbital cortex, dorsolateral orbital cortex, anterior olfactory nucleus (lateral part), anterior olfactory nucleus (posterior part), anterior olfactory nucleus (medial part) and infralimbic cortex, PL with S1, S2, primary auditory cortex, secondary auditory cortex (dorsal area) and AuV, MTL with LENT, PRH, ECT, TeA and dorsal endopiriform nucleus as well as PIR, HIP and AMY with lateral amygdaloid nucleus (ventrolateral part, ventromedial part), basomedial amygdaloid nucleus (posterior part) and basolateral amygdaloid nucleus (posterior part). At the end of the experiment at 504 h p.i., mild to moderate meningoencephalitis was still present in the mes- and telencephalic regions described above. Very mild inflammation in ganglia, BS,

and spinal cord was inconsistently observed at that time. In contrast, no signs of inflammation were found in the mock-infected control mice sacrificed in parallel.

## Brain inflammation and neuronal degeneration

Histopathologically, meningoencephalitis was characterized by infiltration of lymphocytes and histiocytes with marked perivascular cuffing and neuronal degeneration followed by satellitosis, neuronophagy and reactive gliosis. Representative areas of inflammation at different time points in TG, BS and MTL are illustrated in Fig 10. Extensive neuronal necrosis was found in

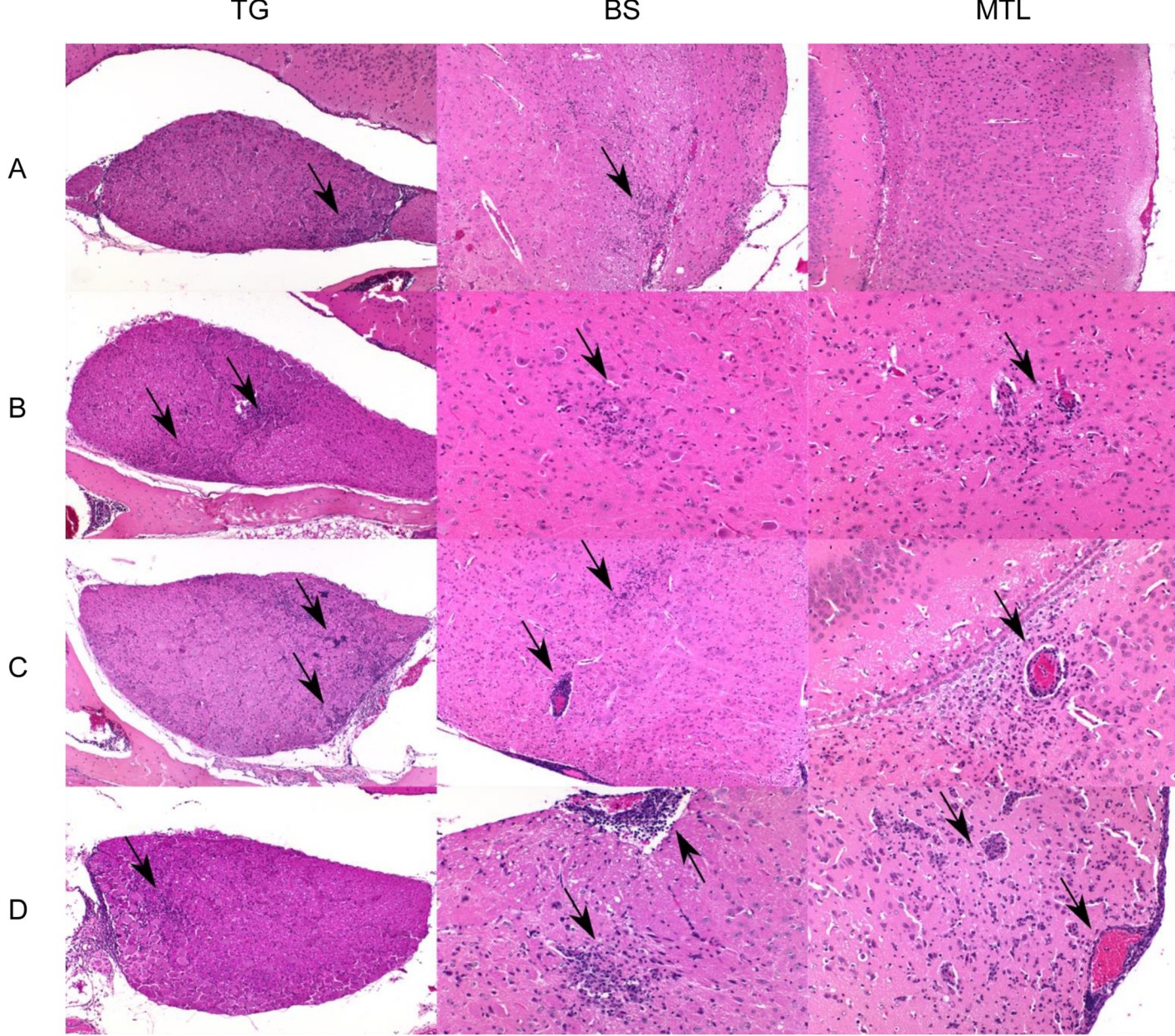

**Fig 10. Detection of inflammation in mice infected with PrV-ΔUL21/US3Δkin.** Hematoxylin and eosin stains of the trigeminal ganglion (TG), brainstem (BS) and mesiotemporal lobe (MTL) are shown in A) 132–168 h p.i., B) 192–240 h p.i., C) 264 h p.i. and D) 288–360 h p.i. Perivascular and parenchymal inflammatory infiltrates consisting of lymphocytes and histiocytes are highlighted by arrows.

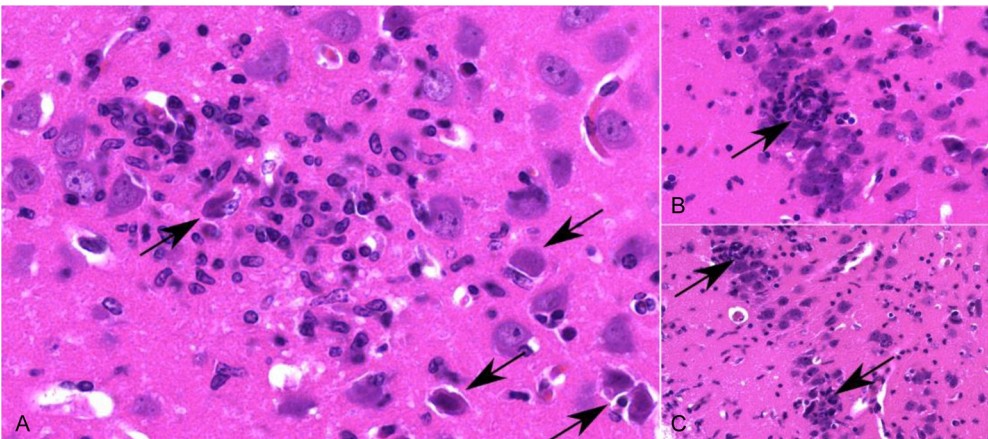

**Fig 11. Inflammatory response in the mesiotemporal lobe.** (A) Necrotic neurons (arrow) are surrounded by inflammatory cells and activated microglia. (B and C) Necrotic neurons in the cerebral cortical layer (arrow) with satellitosis and gliosis, as well as parenchymal infiltration of the adjacent neuropil (C).

the cerebral cortex, which presented as hypereosinophilic, shrunken neurons with surrounding inflammatory cell infiltrates (Fig 11). To identify apoptotic cells, representative sections of the TG, BS and MTL of PrV-ΔUL21/US3Δkin infected mice were stained for active caspase-3. Apoptosis was mainly detected in areas without inflammation and was found in neurons and to a lesser extent in glial cells (S3 Fig). Only single caspase-3-positive cells were present in areas with inflammatory response.

## Discussion

In this study, we characterized a PrV mutant simultaneously lacking the tegument protein pUL21 and the pUS3 kinase activity in cell culture and in mice. A mutant completely lacking both genes (PrV-ΔUL21/US3) was exceptional in previous *in vivo* testing of different PrV mutants [28, 29], since it was the only causing productive infection that most of the infected animals were able to control and survive. This exceptional finding prompted us to investigate infection with this virus mutant in more detail.

To test the influence of the pUS3 kinase activity, we generated a novel mutant, PrV-ΔUL21/US3Δkin, where only the kinase function was inactivated by a point mutation in the catalytic center, leaving the protein otherwise intact. Growth properties of PrV-ΔUS3 and PrV-US3Δkin were comparable as shown previously [40]. PrV-ΔUL21/US3Δkin displayed an approx. 50- to 100-fold titer reduction in cell culture although both proteins, pUL21 and pUS3, are dispensable for viral replication *in vitro* and *in vivo* of PrV [40–42], and HSV-1 [43, 44]. Cell-to-cell spread of PrV-ΔUL21/US3Δkin was significantly impaired and plaque diameters reached only approx. 50%. However, despite these impairments PrV-ΔUL21/US3Δkin was able to replicate productively in cell culture, a prerequisite for the subsequent animal studies. PrV-ΔUL21/US3Δkin was further investigated in our murine intranasal infection model and compared to PrV-Ka and the corresponding single mutants.

Animals infected with PrV-Ka and PrV-US3Δkin showed a similar MTD of 62 h and 63 h, respectively, and neuroinvasion of both viruses was mainly restricted to the trigeminal pathway including first and second order neurons. Mice infected with PrV-ΔUL21 showed an increased MTD of approximately 110 h, accompanied by a more widespread, but still limited invasion of the cerebral cortex. In contrast to PrV-Ka and PrV-US3Δkin infected animals, PrV-ΔUL21-infected mice showed fulminant ganglionitis as well as mild brainstem

encephalitis, but no inflammation in the cerebral cortex where only few neurons were infected. Both gene products, pUL21 and pUS3 are conserved among the *Alphaherpesvirinae* and dispensable for viral replication in PrV as well as in HSV-1 [39, 41, 44, 45]. pUS3 is a multifunctional protein which is involved in different processes during viral infection [46]. Whereas PrV lacking pUS3 is attenuated in pigs [45], only a slight delay in disease progression has been observed in mice [28]. Different mechanisms by which pUS3 modulates pathogenicity have been reported. Escape from the host immune response by downregulation of cell surface receptors including CD1d or major histocompatibility complex class I is induced by pUS3 [47–50] preventing efficient killing of virus-infected cells [48, 50]. pUS3 has been further demonstrated to affect T cell activity [51–53] and interferon signaling followed by impaired immune responses against viral infection resulting in longer survival times. Moreover, pUS3 has been reported to block virus-induced apoptosis by controlling a multitude of apoptosis-associated factors [54–61]. However, mice infected with PrV mutants lacking pUS3 completely or defective in kinase activity showed only a minimal delay in the onset of disease, and clinical signs were similar as in PrV-Ka infected animals. Like in PrV-Ka infected mice only very mild inflammatory infiltration was present in affected ganglia. In contrast, the functional role of pUL21 is still poorly understood. pUL21 has been implicated in different steps of the viral replication cycle in non-neuronal and neuronal cells [39, 62–67]. In particular, mutations in or complete absence of pUL21 affect retrograde transport processes in neurons [68, 69], and point mutations in the UL21 gene of live-attenuated vaccine strain PrV-Bartha contribute to its avirulence in pigs [42]. Repair of the UL21 locus in PrV-Bartha has been shown to enhance retrograde intraaxonal and transneuronal spread [69]. The delay in retrograde transport of UL21-null PrV has been linked to an interaction of the carboxyl terminus of pUL21 with Roadblock-1, a dynein light chain belonging to the dynein motor complex important for retrograde transport along microtubules [68]. Mice infected intraocularly or intranasally with a UL21-deleted PrV showed prolonged survival times compared to mice infected with wildtype virus [68] confirming previous results that pUL21 has an impact on neuroinvasion and spread [28]. As shown in the present study infection with PrV lacking pUL21 resulted in extended times until death compared to PrV-Ka. However, all mice developed severe clinical signs and had to be euthanized, although the disease developed slower. In the light of decelerated axonal spread [68, 69] and subsequent delayed infection of higher areas an immune response is beginning to develop in PrV-ΔUL21-infected mice, but not effective enough to eliminate the virus. In summary, these data confirm that pUL21 contributes to neuroinvasion, but elucidation of the molecular function requires more in-depth investigation.

Severe clinical signs as seen in PrV-Ka and single mutant infected animals seem not to correlate with the presence or absence of inflammation, but rather be associated with brainstem infection, particularly with damage of neurons in the formatio reticularis, which are important to preserve cardiovascular functions. Sudden death of infected neurons of the FR cause cardio-respiratory collapse as observed in our experiment when the animals developed dyspnea, which was also reported after Enterovirus 71 infection of mice leading to severe encephalomyelitis [70].

In contrast to mice infected with PrV-Ka or the single mutants, mice infected with PrV-ΔUL21/US3Δkin survived, except for one animal, which had to be euthanized 239 h p.i. The phenotype observed after simultaneous deletion of pUL21 and pUS3 kinase *in vivo* and *in vitro* might be either based on additive effects of usually separate functions or explained by a functional relationship between these two proteins. In line with this, previous data indicated that in the absence of pUL21 less pUS3 is incorporated into progeny virions [65, 71] which could result in the amplification of effects observed with the single deletion mutants. Aside from the mutations infection with PrV-ΔUL21/US3Δkin mimics herpesviral encephalitis very well in different ways.

Infection with PrV-ΔUL21/US3Δkin caused severe meningoencephalitis with extensive neuronal necrosis, meningeal and perivascular infiltrates of lymphocytes and histiocytes as well as glial activation, which is fully consistent with lesions described for HSE patients [8, 72–75]. Histopathological changes including viral antigen detection and inflammatory response in mice infected with PrV-ΔUL21/US3Δkin were mainly observed unilateral in FL, MTL and PIR as well as in HIP as reported for patients who died of HSE [74, 76, 77]. Apoptotic cells were identified mainly in areas without inflammatory reaction indicating increased neuronal death in the context of virus-induced neuronal damage [78–80]. So far, PrV-ΔUL21/US3Δkin is the only PrV mutant that establishes a productive infection in mice, which the animals are able to control and survive despite widespread neuroinvasion towards the cerebral cortex and a concomitant severe inflammatory response particularly in the frontal and mesiotemporal lobes. While studies in mice showed that HSV-1 rather causes a diffuse inflammation throughout the brain or only in the brainstem depending on the inoculation route used [21–23], in our experiments the typical focal lesions in the mesiotemporal and frontal lobes as observed in human patients were prevalent. Thus, our model authentically reproduces the pattern of herpesviral encephalitis in humans. Since details on the tropism of HSV-1 in mice and humans are still unknown the use of our mouse model offers new perspectives to investigate how alphaherpesviruses gain access to the brain and why infection is established preferably in the temporal and frontal lobes. Although speculative, neurons may differ in their susceptibility to infection, which in turn may result in infection of more vulnerable neuronal subtypes. Unique innate immune responses have been described for granule cell neurons of the cerebellum and cortical cerebral neurons upon West Nile virus infection [81], but little is known in the context of alphaherpesvirus infections [82].

In the present study, we were able to trace the immune response against herpesvirus infection over a long period. Due to the extended survival time of mice infected with PrV-ΔUL21/US3Δkin, it will be feasible to investigate long-term pathobiological mechanisms for development and during herpesviral encephalitis, their contribution to disease severity and associated immunopathological processes as described for HSE [83–85].This was not possible in mice infected with virulent HSV-1, regardless of intranasal or intracerebral infection [86–89], since animals invariably succumbed to the infection.

Behavioral changes including stargazing which occurred with increasing inflammation in the brain indicate impairments, which represent analogies to neurophysiological phenomena in HSE or VZV patients, which are mostly related to damages in the limbic system including HIP [90, 91]. Behavioral as well cognitive impairment may well be even more pronounced beyond the 21 days of our infection experiment, where all mice still showed brain inflammation and exhibited signs of abnormality. Problems with memory and cognitive failure as well as development of epilepsy have been discussed in association with HSV-1 infection [92]. Whether mice infected with PrV-ΔUL21/US3Δkin develop similar clinical signs remains to be tested in long-term studies and analyzed by appropriate learning and memory tests. However, to our knowledge, this is the first description of behavioral failure in mice in the course of herpesviral encephalitis, which offers new perspectives to study behavioral disorders associated with herpesviral infection.

Herpesviruses establish latency after infection and viral DNA was found not only in the peripheral nervous system, but also in the CNS in HSE patients [93, 94]. In our study, viral antigen was not detectable by immunohistochemistry beyond 360 h p.i., which might be due to the low amount of viral antigen in the tissue section analyzed [95] or explained by early virus clearance leading to latent infection. This parallels HSE where viral antigen could be detected only within 3 weeks of onset of the disease in HSE patients, but was absent afterwards [74]. Experiments to test for latent DNA of PrV-ΔUL21/US3Δkin are planned for the future.

In HSE patients, involvement of BS as seen in mice infected with PrV-ΔUL21/US3Δkin has been reported but thought to be a special form of manifestation of HSV-1 encephalitis [96–98]. Involvement of BS may point to the trigeminal pathway of neuroinvasion. However, in PrV-ΔUL21/US3Δkin infected mice viral antigen and brain inflammatory response were first present in cortical areas such as MTL and PIR, which receive projections from the olfactory bulb indicating the possibility of infection via the olfactory route as discussed in the development of HSE.

In mice infected with PrV-ΔUL21/US3Δkin, myelitis as well as zosteriform alopecic skin lesions which are typical pathologic hallmarks of infection primarily with VZV [10, 99] were observed. In VZV, segmental rash occurs during virus reactivation in dorsal root ganglia, which are associated with a corresponding dermatome, but, in contrast to our finding, carry viral antigen [10, 100]. The pathogenesis of myelitis is still unknown, but it is suspected that the virus spreads from infected dorsal root ganglia to the spinal cord [10]. However, our data also indicate that PrV-ΔUL21/US3Δkin may has been transported along the spinal cord resulting in lesions distant from the head. In this respect, virus-induced lesions in peripheral and central sensory tracts may be the reason for the pruritus observed in different body regions of mice after infection with PrV-ΔUL21/US3Δkin as well as in herpesvirus-infected humans. As reported in humans the trigeminal trophic syndrome (TTS) caused by herpesviral infections also results in painless itch of the facial skin caused by injury of the trigeminal nerve, TG and even parts of the brain including TH [101–103].

Ganglionitis as well as ganglioneuritis as seen in our model are usually not observed in patients suffering from HSV-1 associated encephalitis, except for one case report of cervical dorsal root ganglionitis without encephalitis [104]. However, for VZV patients such cases have been repeatedly described [16, 105, 106]. For HSV-1 and VZV, viral genomes have been detected independently in sensory and autonomic ganglia (e.g. SCG) of the human head and neck [11].

In summary, mice infected with PrV lacking pUL21 and functional pUS3 kinase were able to survive infection despite extensive neuroinvasion and severe meningoencephalitis. The animal model presented here reflects important aspects of herpesviral encephalitis in humans including the characteristic distribution of histopathological changes and behavioral abnormalities. Thus, we suggest that this animal model is highly suitable for further investigations towards understanding the pathogenesis of herpesviral encephalitis.

## Material and methods

### Viruses and cells

All virus mutants used were derived from PrV wildtype strain Kaplan (PrV-Ka) [107]. PrV-Ka, PrV-ΔUL21 [39] and PrV-US3Δkin [40] were propagated in rabbit kidney cells (RK13), an established cell line for PrV infection in our lab, and grown at 37°C in minimum essential medium (MEM) (Invitrogen) supplemented with 10% fetal calf serum.

### Generation of PrV-ΔUL21/US3Δkin

PrV-ΔUL21/ΔUS3gfp lacking US3-specific sequences but expressing green-fluorescent protein was generated after co-transfection of PrV-ΔUL21 viral DNA [39] with plasmid pΔUS3gfpII [41] into RK13 cells. Green fluorescing plaques were purified to homogeneity and absence of US3 was verified by Southern and Western blotting. PrV-ΔUL21/US3Δkin was isolated after co-transfection of plasmid pUC-US3Δkin [40] with PrV-ΔUL21/US3gfp genomic DNA and purification of non-fluorescing plaques. Immunofluorescence and immunoblotting using a rabbit anti-pUS3 (1:50 000) [41] and anti-pUL21 (1:20 000) [39] serum were used to verify the

absence of the UL21 protein and presence of pUS3. Presence of the mutation in the catalytic domain of pUS3 was confirmed by sequence analysis.

## In vitro replication studies

One-step growth kinetics were established by infection of RK13 cells with PrV-Ka, PrV-US3Δkin, PrV-ΔUL21, and PrV-ΔUL21/US3Δkin at a multiplicity of infection (MOI) of 3 as described [108]. Plaque assays were used to determine the titer of virus progeny on RK13 cells. Three independent experiments were performed to assess the mean viral titers. In addition, plaque diameters of 20 plaques per virus in three independent assays were measured to determine characteristics of cell-to-cell spread of the respective virus mutants.

## Ethics statement

Animal experiments were approved by the State Office for Agriculture, Food Safety and Fishery in Mecklenburg-Western Pomerania (LALFF M-V) with reference number 7221.3-1-064/17.

## Animal experiments

In the present study, we used 6–8 weeks old female CD-1 mice as our standardized animal model for PrV infection [27–29]. Female animals were used to facilitate group housing. The animals were purchased from Charles River Laboratories and housed in groups of maximal five animals per cage at the animal facilities of the Friedrich-Loeffler-Institut, Greifswald-Insel Riems. Animals were kept under controlled conditions of 12 h light: 12 h dark with free access to food and water. After one week of acclimatization, mice were deeply anesthetized with 200 µl of a mixture of ketamine (60mg/kg) and xylazine (3mg/kg) dissolved in 0.9% sodium chloride, which was administered intraperitoneally. Afterwards a total volume of 5 µl each of the corresponding virus suspension was inoculated in both nostrils (1x10$^4$ plaque forming units). Mock mice were treated with cell culture supernatant from RK13 cells (MEM + 5% FCS) accordingly.

**Determination of the mean time to death and clinical evaluation.** Mean time to death (MTD) was established after inoculation of mice with PrV-Ka (n = 4), PrV-US3Δkin (n = 4), PrV-ΔUL21 (n = 4) and PrV-ΔUL21/US3Δkin (n = 6) and cell culture supernatant (n = 2) as control. Following intranasal inoculation, animals were monitored twice daily every 12 hours over a period of maximal 21 days. Mice were evaluated based on a predefined scoring system with the following three categories: (I) external appearance, (II) behavior and activity and (III) body weight (S1 Table). For each category, a score ranging from 0 to 3 was determined. Score 3 in one category or score 2 in all three categories was defined as the humane endpoint on which the animal was euthanized.

**Determination of the kinetics of viral spread and inflammatory changes.** To assess the kinetics of viral spread and inflammatory changes after inoculation with PrV-Ka, PrV-US3Δkin, PrV-ΔUL21 and PrV-ΔUL21/US3Δkin a minimum of three mice was sacrificed and examined at each time point. For analysis of PrV-Ka and PrV-US3Δkin animals were sacrificed after inoculation at the following time points: 24, 36, 48, 60 h. For investigation of PrV-ΔUL21 mice were sacrificed at 24, 36, 48, 60, 72, 84, 96, 108 and 120 h. Animals inoculated with PrV-ΔUL21/US3Δkin were sacrificed at 24, 36, 48, 60, 72, 84, 96, 108, 120, 132, 144, 156, 168, 192, 216, 240, 264, 288, 312, 336, 360, 432 and 504 h. After 24, 168, 312 and 504 h one mock-infected mouse each was sacrificed.

**Euthanasia and tissue sample collection.** Animals were deeply anesthetized with isoflurane, and subsequently sacrificed through cardiac bleeding with a 21-gauge needle. The head

was removed at the level of the first or second cervical vertebrae. The skull was taken off to allow accurate fixation and subsequent evaluation of all anatomical-physiological structures of the head of all mice. In addition, the spinal column including the spinal cord as well as affected skin tissue were collected from several mice. All tissue specimens were fixed in 10% neutral buffered formalin for at least one week.

**Histopathology and immunohistochemistry.** Following fixation, heads and spinal column were decalcified for at least three days in Formical 2000 (Decal, Tallman, N.Y.). Heads were subsequently sliced from caudal to cranial to obtain eight coronal sections of 2–3 mm thickness each. The spinal column was cut into 2–3 mm sections at the cervical (3–4 sections), thoracical (7–8 sections), lumbal (4–5 sections), and sacral (3 sections) level. Tissue specimen of the head, spinal cord and skin were then embedded in paraffin wax and cut at 3 μm thick slices. Two slices per each head section separated by 100 μm were collected to increase the amount of tissue for investigation. Therefore, 16 head sections of each animal were analyzed histopathologically and immunohistochemically, respectively. For light microscopical investigation, the sections were mounted on Super-Frost-Plus-Slides (Carl Roth GmbH, Karlsruhe, Germany) and stained with hematoxylin-eosin.

For immunohistochemistry, sections were dewaxed and rehydrated. Intrinsic peroxidase activity was blocked through 3% of hydrogen peroxide (Merck, Darmstadt, Germany) treatment for 10 min. Sections were then incubated with undiluted normal goat serum for 30 min to block unspecific binding sites before incubation with the respective primary antibody. After washing with Tris-buffered saline (TBS) the rabbit glycoprotein B-specific antiserum (1:2000, diluted in TBS, 60 min, [109]) was used to detect infected cells. To investigate apoptosis a rabbit antiserum against active caspase-3 (Promega, Madison, USA; 1:200, diluted in TBS) was used. Sections were rinsed and subsequently incubated with a biotinylated goat anti-rabbit IgG (Vector Laboratories, Burlingame, CA; diluted 1:200 in TBS, 30 min), followed by an incubation with ABC (Vector) diluted 1:10 in TBS for 30 min, providing the conjugated horseradish peroxidase. Positive reactions were visualized with AEC-substrate (DAKO, Hamburg, Germany). After washing with deionized water, the sections were counterstained with Mayer's Hemalaun for 2 min and mounted with Aquatex (Merck). To verify positive viral antigen staining, selected sections were treated with a rabbit antiserum against the major capsid protein pUL19 [110] and further examined with RNAScope using a probe against the same epitope. Comparable results were obtained for all staining techniques.

**Semi quantitative analysis of histological specimens and scoring of viral antigen distribution.** Light microscopical examination was conducted using a Zeiss Axio Scope.A1 microscope equipped with 5x, 10x, 20x, and 40x N-ACHROPLAN objectives (Carl Zeiss Microscopy GmbH, Jena, Germany).

Each of the immunohistochemically stained head and spinal cord as well as skin sections were examined for positive anti-PrV-glycoprotein B reactions. For identification of viral antigen positive neural structures in the brain the mouse brain atlas was used, and distribution of viral antigen positive cells in the respective sections were scored as follows: 0 = negative, 1 = focal to oligofocal, 2 = multifocal, 3 = diffuse. Points given for each section were added up to obtain a total sum for the whole head, which was then taken into analysis. Since each of the eight head main sections was investigated twice, the total amount of the first sectional plane and the second sectional plane, respectively, was averaged and statistically analyzed.

**Scoring of inflammation.** Tissue sections were stained with hematoxylin and eosin, investigated for histopathological changes and analyzed as described above. Evaluation was based on the following scoring system: 0 = no inflammation, 1 = mild inflammation, 2 = moderate inflammation, 3 = severe inflammation. Along with the inflammatory response, infiltration of immune cells as well as reactive changes (e.g. neuronal necrosis, degeneration and loss,

gliosis, satellitosis, neuronophagy) were additionally characterized at each day of investigation. Total values were analyzed as described above (see scoring of viral antigen distribution).

**Statistical analysis.** Statistical analysis and graphical presentation were performed by using GraphPad Prism. A Kaplan-Meier curve was generated to illustrate the relative survival rate after inoculation with the respective viruses followed by survival analysis using the log-rank test. All values were analyzed using the standard error of the mean (SEM). All animal groups examined on 24, 36, 48, 60, 72, 84, 96, 108, 120, 132, 144, 156, 168, 192, 216, 240, 264, 288, 312, 336, 360, 432 and 504 h compared to mock-infected mice (control) were analyzed by the nonparametric Kruskal-Wallis test followed by pairwise uncorrected Dunn's *post hoc* tests. Growth of virus mutants in cell culture were analyzed with an ordinary one-way ANOVA followed by Dunnet's multiple comparison test. Average values and standard deviations from three independent experiments were calculated, and the mean of each virus mutant and time point was compared to PrV-Ka as control. P-values with a significance limit of $\leq 0.05$ were considered and indicated by an asterisk ($^*$) in the graphs.

# Supporting information

**S1 Table. Scoring system for mice infected with PrV.** Three categories including (I) external appearance, (II) behavior and activity and (III) body weight were assessed daily and utilized to group mice into either mildly (max. score 1 in three out of three categories), moderate (max. score 2 in two out of three categories) and severely affected (max. score 2 in all categories or max. score 3 in one out of three categories).
(DOCX)

**S2 Table. Clinical signs and mean time to death (MTD).** The onset of disease, MTD as well as typical clinical signs, which have been observed in mice infected with different virus mutants are listed.— = not present, ✓ = present, (✓) = occasionally observed.
(DOCX)

**S1 Fig. Viral antigen distribution in infected with PrV-Ka.** Immunohistochemistry using an anti-glycoprotein-B-antibody was performed to detect PrV in the respective tissues. A) nasal respiratory epithelium (RE), B) trigeminal ganglion (TG), C) spinal trigeminal nucleus (Sp5), D) nasopharynx (NP), E) salivary glands (SG), F) pterygopalatine ganglion (PtG), G) superior cervical ganglion (SCG) and H) cerebral cortex. Viral antigen has not been detected in the cerebral cortex.
(TIF)

**S2 Fig. Schematic illustration of spread of PrV-ΔUL21/US3Δkin in mice after intranasal infection.** Localization of viral antigen detection is shown for different times p.i. No viral antigen was detectable later than 360 h p.i. RE = respiratory epithelium, NG = nasal gland, OM = oral mucosa, NP = nasopharynx, SG = salivary gland, TG = trigeminal ganglion, PtG = pterygopalatine ganglion, SCG = superior cervical ganglion, Cu = cuneate nucleus, Sol = solitary tract, Sp5 = spinal trigeminal nucleus, FR = reticular formation, IO = inferior olive, LPBC = lateral parabrachial nucleus, Pr5 = principal sensory trigeminal nucleus, Me5 = mesencephalic trigeminal nucleus, PAG = periaqueductal grey, VTA = ventral tegmental area, TH = thalamus, HYPO = hypothalamus, OB = olfactory bulb, FL = frontal lobe, PL = parietal lobe, PIR = piriform lobe, MTL = mesiotemporal lobe, HIP = hippocampus, AMY = amygdala.
(TIF)

**S3 Fig. Immunohistochemistry showing active caspase-3.** The trigeminal ganglion (TG), brainstem (BS) and mesiotemporal lobe (MTL) were examined for apoptosis. A, C and E)

Caspase-3-positive cells only appeared occasionally in regions adjacent to inflammatory spots (asterisk) and included both neuronal and glial cells (arrowhead). B, D and F) Apoptosis of neurons or glial cells was mainly detectable in non-inflamed areas which are indicated by arrows.
(TIF)

## Acknowledgments

For expert technical assistance we thank Silvia Schuparis, Gabriele Czerwinski, and Cindy Krüper.

## Author Contributions

**Conceptualization:** Julia Sehl, Thomas C. Mettenleiter.

**Data curation:** Julia Sehl.

**Formal analysis:** Julia Sehl, Julia E. Hölper.

**Funding acquisition:** Julia Sehl, Jens P. Teifke, Thomas C. Mettenleiter.

**Investigation:** Julia Sehl, Julia E. Hölper, Barbara G. Klupp, Christina Baumbach.

**Methodology:** Julia Sehl, Barbara G. Klupp, Jens P. Teifke, Thomas C. Mettenleiter.

**Project administration:** Jens P. Teifke, Thomas C. Mettenleiter.

**Resources:** Jens P. Teifke, Thomas C. Mettenleiter.

**Supervision:** Jens P. Teifke, Thomas C. Mettenleiter.

**Validation:** Julia Sehl, Barbara G. Klupp, Jens P. Teifke, Thomas C. Mettenleiter.

**Visualization:** Julia Sehl.

**Writing – original draft:** Julia Sehl.

**Writing – review & editing:** Barbara G. Klupp, Jens P. Teifke, Thomas C. Mettenleiter.

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
