## [Decision Letter · Decision Letter 0]

10 Feb 2020

Dear Prof. Dr. Thomas,

Thank you very much for submitting your manuscript "A Novel Animal Model for Herpesviral Encephalitis in Humans" for consideration at PLOS Pathogens. As with all papers reviewed by the journal, your manuscript was reviewed by members of the editorial board and by several independent reviewers. In light of the reviews (below this email), we would like to invite the resubmission of a significantly-revised version that takes into account the reviewers' comments.

All reviewers indicate that this model has some unique properties and value. Reviewers 1 and 2 recommend major revision and provide the reasons for their recommendation. Reviewer #3 recommends reject.

While a number of comments are provided, the primary underlying concern is that the authors do not provide mechanistic insight: how do the mutation in this PRV mutant lead to this altered host response? The secondary concern is presented by reviewer #3 who questions if this model will be useful for understanding pathogenesis of HSV and VZV. Without a mechanistic understanding, is this altered host response relevant at all to HSV and VZV?

The authors must deal with the major concerns expressed by the two major issue comments of reviewer #1. I don't think you need to do TEM of the mutant particles. In addition, all the comments of reviewer #2 must be addressed as they relate to providing some insight into mechanism. The concerns of Reviewer #3 are important but deal primarily with the assertions of the models relevance. These concerns can be addressed by rewriting and trying to relate the phenotypes of PRV mutants to HSV and VZV brain infections.

Regarding mechanism, there is a paper on the role of PRV UL21 and neuronal infection: Curonovic et al (repair of the UL21 locus in PRV Bartha enhances the kinetics of retrograde trans neuronal infection in vitro and in vivo. JVI 2009. 83:1173)

Overall, the manuscript needs rewriting focusing more on mechanism and relevance of this mechanism to the pathogenesis of two human alphaherpesviruses. Many of the minor issues that were presented also should be addressed for clarification.

We cannot make any decision about publication until we have seen the revised manuscript and your response to the reviewers' comments. Your revised manuscript is also likely to be sent to reviewers for further evaluation.

Sincerely,

Lynn W. Enquist

Guest Editor

PLOS Pathogens

Blossom Damania

Section Editor

PLOS Pathogens

Kasturi Haldar

Editor-in-Chief

PLOS Pathogens

orcid.org/0000-0001-5065-158X

Michael Malim

Editor-in-Chief

PLOS Pathogens

orcid.org/0000-0002-7699-2064

All reviewers indicate that this model has some unique properties and value. Reviewers 1 and 2 recommend major revision and provide the reasons for their recommendation. Reviewer #3 recommends reject.

While a number of comments are provided, the primary underlying concern is that the authors do not provide mechanistic insight: how do the mutation in this PRV mutant lead to this altered host response? The secondary concern is presented by reviewer #3 who questions if this model will be useful for understanding pathogenesis of HSV and VZV. Without a mechanistic understanding, is this altered host response relevant at all to HSV and VZV?

The authors must deal with the major concerns expressed by the two major issue comments of reviewer #1. I don't think you need to do TEM of the mutant particles. In addition, all the comments of reviewer #2 must be addressed as they relate to providing some insight into mechanism. The concerns of Reviewer #3 are important but deal primarily with the assertions of the models relevance. These concerns can be addressed by rewriting and trying to relate the phenotypes of PRV mutants to HSV and VZV brain infections.

Regarding mechanism, there is a paper on the role of PRV UL21 and neuronal infection: Curonovic et al (repair of the UL21 locus in PRV Bartha enhances the kinetics of retrograde trans neuronal infection in vitro and in vivo. JVI 2009. 83:1173)

Overall, the manuscript needs rewriting focusing more on mechanism and relevance of this mechanism to the pathogenesis of two human alphaherpesviruses. Many of the minor issues that were presented also should be addressed for clarification.

Reviewer's Responses to Questions

**Part I - Summary**

Reviewer #1: General comment. The Mettenleiter lab presents a compelling argument, reinforced with data from a UL21/US3 mutated PRV strain, that PRV should be considered (or reconsidered) as an animal model system for investigation of HSV or VZV induced neurological disease. This reviewer agrees with their basic premise about the potential value of using PRV as a model virus to study neurological mechanisms of other alpha herpes viruses. Several comments request additional information or more citations of research by other herpes labs. In the most important comment, a request is made for more data to be included within an expanded Figure 7. In general, this manuscript will be of broad interest to alpha herpes virologists.

Reviewer #2: For this study, the authors characterized a PRV mutant that has the kinase domain of US3 mutated and UL21 deleted (PRV-∆UL21/US3∆kin). This double mutant grows less efficiently in rabbit kidney cells relative to wt Kaplan or the single mutant viruses. Surprisingly, the PRV-∆UL21/US3∆kin mutant does not kill CD-1 mice during acute infection. Conversely, CD-1 mice infected with wt PRV (Kaplin strain) or PRV mutants in UL21 or the US3 kinase domain do not recover from acute infection. However, PRV-∆UL21/US3∆kin caused encephalitis following infection via the nasal route, and neuro-invasion occurred along the trigeminal pathway. Furthermore, PRV-∆UL21/US3∆kin virus spread extensively in the CNS and caused extensive inflammation in the brainstem. Interestingly, most mice infected with PRV-∆UL21/US3∆kin showed behavioral abnormality and slow movement, which is analogous to symptoms of encephalitis in humans. In general, this is an intriguing study that provides a new model to study HSE and test new drugs. With that said, there are several issues that need to be addressed, as summarized below.

Reviewer #3: The original work in this manuscript involves the initial characterization in mice of the pathogenesis of a pseudorabies virus (PRV) mutant with an engineered deletion that removes the pUL21 tegument protein AND the kinase domain of the US3 protein. The authors suggest that pattern of neuropathogenesis may be useful in understanding the pathogenesis of neurotropic human alphaherpesviruses including VZV and HSV.

Weaknesses:

(1) It is exceedingly unlikely that there will be adequate identity in the pathological processes from tropism to spread to immune responses between this engineered PRV and HSV or VZV that this will in fact be a particularly useful model for understanding pathogenesis of those viruses or that results will be directly translatable. In the case of HSV several murine models do already exist and in the case of VZV this does mnot appear to mimic the human vasculopathy that is central to the "encephalitis".

(2) The most interesting observation is that the dbl mutant PRV produces both an extensive pattern of neuronal injury/neuroinvasion but also allows increased survival. Unfortunately no mechanistic insights are provided into these processes. The authors seem aware of this (line 408 et seq. on p.23):" investigation of pathobiological mechanisms as well as the nature and role of the immune control of this life threatening disease is feasible"- but don't actually provide any insights in this area. It is this type of insight that would make for an impactful paper. How does the double mutant cause the effects seen? Do these mice show long term cognitive or other sequelae that would add to the uniqueness of this model?

Strengths:

Although wt PRV has been extensively studied in murine models, this mutant clearly has some unique properties. Understanding the mechanisms by which these occur would be very interesting. Most if not all the observations are "phenomenological"- e.g. the dbl mutant grows less well on RK13 cells and makes small plaques, its is substantially less lethal after intranasal inoculation than either sgl mutant UL21 or US3 PRV or wt PRV (Ka strain). it shows a distinct pattern of brain antigen distribution, inflammation, .etc. The critical additions that would add impact include on the viral side understanding how the dual deletions contribute to the phenotype and on the host side the differences in immune response or other factors that contribute to the altered pathogenesis and injury with lessened mortality.

**Part II – Major Issues: Key Experiments Required for Acceptance**

Reviewer #1: 1. Introduction. Lines 151-177 and reference 50. This long paragraph in the introduction describes all the Results. This paragraph could be placed in the Discussion section. Reference 50 in this section is important because it cites a paper that found UL21 to be essential for the entry of viruses into nerves. In the Discussion, the authors should describe reference 50 more completely and they also need to describe the consequence of removal of a tegument protein needed for neuroinvasion in a mutated virus that they propose as a model virus to study viral encephalitis. In lines 408-410, the authors state that long term investigation of the PRV pathobiological mechanism is possible due to the extended survival times after infection with the double-mutant virus. Yet, isn’t the extended survival time due to the fact that a neurotropic protein UL21 has been removed? If HSV and VZV have the homolog, can the viruses be compared to PRV without UL21?

2. Results, line 302, Figure 7. Based on the data presented in Results, the light micrographs in Figure 7 do not seem to represent the same degree of infection. The infectious focus is very small. Why use only one anti-PRV antibody? Instead, why not use a cocktail of perhaps 3 antibodies against PRV? Perhaps gB, plus major capsid protein plus a tegument protein. Also, show us at least one panel at a higher magnification so that we can determine if the positive gB focus represents a single cell or a small cluster of infected cells, or syncytium?

Also, we do not know the effect of removal of a tegument protein (UL21) on the structure of the mutant virion after secondary assembly or the amount of gB in the envelope of that virion. This group is also known for their electron microscopy of viral particles. Why not include a few electron micrographs to document the presence of virions in the ganglia and brain? Electron micrographs are also important because the authors document a major decrease in titer of their VL21/US3 mutant virus in Figure 2. Perhaps show a representative side-by-side TEM comparison of wild-type PRV and mutant PRV in the nucleus of an infected cell and in the cytoplasm of an infected cell. The US3 mutation may alter egress from nucleus and the UL21 mutation may alter secondary envelopment. Prediction: there will be a large decrease in percentage of prototypical virions in cytoplasm.

Reviewer #2: 1. For the mean time to death, 4 mice were used for the Kaplan strain and the single mutants. Six mice were infected with PRV-∆UL21/US3∆kin. This does not seem enough mice for these studies, as there is frequently variability in mouse studies. Was a power analysis performed to estimate how many mice should be used? Also, statistical analysis of the results shown in Figure 3 should be performed.

2. Was the PRV-∆UL21/US3∆kin rescued? If so, this rescued virus should have been used as a control for second site mutations. If PRV-∆UL21/US3∆kin strain was sequenced to confirm there were no additional mutations relative to the parental wt Kaplan strain this should have been stated.

3. While the authors nicely demonstrated viral antigen staining and inflammation in TG, brain stem, and additional areas of the CNS, the same sections should be examined programmed cell death, TUNEL for example. This information would strengthen the case for encephalitis correlating with increased neuronal death.

Reviewer #3: See comments above

Defining how the mutations alter pathogenesis- eg do they impact growth in neuronal tissues/cells, axon transport, host innate or adaptive immunity would be impactful

**Part III – Minor Issues: Editorial and Data Presentation Modifications**

Reviewer #1: 1. Title. The title may be confusing to Readers. To this reviewer, the three words “novel animal model” implies a novel animal, for example, a marmoset or some exotic simian, not a female CD-1 mouse. Also, the same lab used the same word “novel” to describe a new virus found in carp in 2013. See Journals of Fish Diseases 2013. Suggest selection of any alternative word other than novel in their title.

2. Authors summary. Lines 57-58. What is the evidence that VZV is the second most common infection cause of encephalitis? There was a large series of 432 encephalitis cases with documentation by brain biopsy, published by Richard Whitley et al, JAMA 262: 234, 1989. HSV was the most common. There were 10 other viruses identified from encephalitis cases, but none were VZV.

3. Lines 84-85. It is doubtful that VZV is one of the most common viruses causing CNS infections. At least cite the reference by Whitley et al (Comment 4).

4. Line 86. The authors should cite an extremely informative study that determined many cites of HSV and VZV latency in almost all the ganglia found in the head and neck region. The article is from the Atherton lab and was published in the Journal of Infectious Diseases 200: 1901, 2009. In particular, this paper showed both HSV and VZV genomes in the superior cervical ganglion. This ganglion is included in Figure 1 of this manuscript.

.

5. Methods, line 506. Add a few sentences to explain why female CD-1 mice were selected as the experimental animal of choice for this new model of PRV infection. Other murine strains have been used for PRV research by other PRV labs?

Reviewer #2: 1. For Figure 1, the authors used rabbit kidney cells. Since animal studies were performed in mice, why were mouse cells not used?

2. For PRV Kaplan infections, CD-1 mice appear to be very sensitive to death following infection: this clearly is not the case for human infections with virulent field strains of HSV-1. Are there mouse strains that are more resistant to PRV Kaplan? If so, I am curious whether PRV-∆UL21/US3∆kin would have behaved in a mouse strain that was less resistant to PRV-induced death during acute infection?

Reviewer #3: The sections on HSV and VZV are way too long given this is a different virus and the parallels are often strained and unconvincing. Please focus on the specific novel contributions and observations in this study.

PLOS authors have the option to publish the peer review history of their article (what does this mean?). If published, this will include your full peer review and any attached files.

Reviewer #1: No

Reviewer #2: No

Reviewer #3: No
---

## [Editor Report · Decision Letter 1]

29 Feb 2020

Dear Prof. Dr. Thomas,

We are pleased to inform you that your manuscript 'An Improved Animal Model for Herpesviral Encephalitis in Humans' has been provisionally accepted for publication in PLOS Pathogens.

Best regards,

Lynn W. Enquist

Guest Editor

PLOS Pathogens

Blossom Damania

Section Editor

PLOS Pathogens

Kasturi Haldar

Editor-in-Chief

PLOS Pathogens

orcid.org/0000-0001-5065-158X

Michael Malim

Editor-in-Chief

PLOS Pathogens

orcid.org/0000-0002-7699-2064

Thank you for your attention to the reviewers comments.
---

## [Editor Report · Acceptance letter]

20 Mar 2020

Dear Prof. Dr. Mettenleiter,

We are delighted to inform you that your manuscript, "An Improved Animal Model for Herpesviral Encephalitis in Humans," has been formally accepted for publication in PLOS Pathogens.

Best regards,

Kasturi Haldar

Editor-in-Chief

PLOS Pathogens

orcid.org/0000-0001-5065-158X

Michael Malim

Editor-in-Chief

PLOS Pathogens

orcid.org/0000-0002-7699-2064